# TIGHTER BOUNDS LEAD TO IMPROVED CLASSIFIERS

**Nicolas Le Roux**
Criteo Research
`nicolas@le-roux.name`

## ABSTRACT

The standard approach to supervised classification involves the minimization of a log-loss as an upper bound to the classification error. While this is a tight bound early on in the optimization, it overemphasizes the influence of incorrectly classified examples far from the decision boundary. Updating the upper bound during the optimization leads to improved classification rates while transforming the learning into a sequence of minimization problems. In addition, in the context where the classifier is part of a larger system, this modification makes it possible to link the performance of the classifier to that of the whole system, allowing the seamless introduction of external constraints.

## 1 INTRODUCTION

Classification aims at mapping inputs $X \in \mathcal{X}$ to one or several classes $y \in \mathcal{Y}$. For instance, in object categorization, $\mathcal{X}$ will be the set of images depicting an object, usually represented by the RGB values of each of their pixels, and $\mathcal{Y}$ will be a set of object classes, such as "car" or "dog".

We shall assume we are given a training set comprised of $N$ independent and identically distributed labeled pairs $(X_i, y_i)$. The standard approach to solve the problem is to define a parameterized class of functions $p(y|X, \theta)$ indexed by $\theta$ and to find the parameter $\theta^*$ which minimizes the log-loss, i.e.

$$\theta^* = \arg\min_{\theta} -\frac{1}{N} \sum_i \log p(y_i|X_i, \theta) \tag{1.1}$$
$$= \arg\min_{\theta} L_{\log}(\theta) \; ,$$

with

$$L_{\log}(\theta) = -\frac{1}{N} \sum_i \log p(y_i|X_i, \theta) \; . \tag{1.2}$$

One justification for minimizing $L_{\log}(\theta)$ is that $\theta^*$ is the maximum likelihood estimator, i.e. the parameter which maximizes

$$\theta^* = \arg\max_{\theta} p(\mathcal{D}|\theta)$$
$$= \arg\max_{\theta} \prod_i p(y_i|X_i, \theta) \; .$$

There is another reason to use Eq. 1.1. Indeed, the goal we are interested in is minimizing the classification error. If we assume that our classifiers are stochastic and outputs a class according to $p(y_i|X_i, \theta)$, then the expected classification error is the probability of choosing the incorrect class[a]. This translates to

$$L(\theta) = \frac{1}{N} \sum_i (1 - p(y_i|X_i, \theta))$$
$$= 1 - \frac{1}{N} \sum_i p(y_i|X_i, \theta) \; . \tag{1.3}$$

---

[a]In practice, we choose the class deterministically and output $\arg\max_y p(y|X_i, \theta)$.

This is a highly nonconvex function of $\theta$, which makes its minimization difficult. However, we have

$$
\begin{aligned}
L(\theta) &= 1 - \frac{1}{N} \sum_i p(y_i | X_i, \theta) \\
&\leq 1 - \frac{1}{N} \sum_i \frac{1}{K} \left( 1 + \log p(y_i | X_i, \theta) + \log K \right) \\
&= \frac{(K - 1 - \log K)}{K} + \frac{L_{\log}(\theta)}{K} \ ,
\end{aligned}
$$

where $K = |\mathcal{Y}|$ is the number of classes (assumed finite), using the fact that, for every nonnegative $t$, we have $t \geq 1 + \log t$. Thus, minimizing $L_{\log}(\theta)$ is equivalent to minimizing an upper bound of $L(\theta)$. Further, this bound is tight when $p(y_i | X_i, \theta) = \frac{1}{K}$ for all $y_i$. As a model with randomly initialized parameters will assign probabilities close to $1/K$ to each class, it makes sense to minimize $L_{\log}(\theta)$ rather than $L(\theta)$ early on in the optimization.

However, this bound becomes looser as $\theta$ moves away from its initial value. In particular, poorly classified examples, for which $p(y_i | X_i, \theta)$ is close to 0, have a strong influence on the gradient of $L_{\log}(\theta)$ despite having very little influence on the gradient of $L(\theta)$. The model will thus waste capacity trying to bring these examples closer to the decision boundary rather than correctly classifying those already close to the boundary. This will be especially noticeable when the model has limited capacity, i.e. in the underfitting setting.

Section 2 proposes a tighter bound of the classification error as well as an iterative scheme to easily optimize it. Section 3 experiments this iterative scheme using generalized linear models over a variety of datasets to estimate its impact. Section 4 then proposes a link between supervised learning and reinforcement learning, revisiting common techniques in a new light. Finally, Section 5 concludes and proposes future directions.

## 2 TIGHTER BOUNDS ON THE CLASSIFICATION ERROR

We now present a general class of upper bounds of the classification error which will prove useful when the model is far from its initialization.

**Lemma 1.** *Let*

$$
p_\nu(y | X, \theta) = p(y | X, \nu) \left( 1 + \log \frac{p(y | X, \theta)}{p(y | X, \nu)} \right) \tag{2.1}
$$

*with $\nu$ any value of the parameters. Then we have*

$$
p_\nu(y | X, \theta) \leq p(y | X, \theta) \ . \tag{2.2}
$$

*Further, if $\nu = \theta$, we have*

$$
p_\theta(y | X, \theta) = p(y | X, \theta) \ , \tag{2.3}
$$

$$
\left. \frac{\partial p_\nu(y | X, \theta)}{\partial \theta} \right|_{\nu = \theta} = \frac{\partial p(y | X, \theta)}{\partial \theta} \ . \tag{2.4}
$$

*Proof.*

$$
\begin{aligned}
p(y | X, \theta) &= p(y | X, \nu) \frac{p(y | X, \theta)}{p(y | X, \nu)} \\
&\geq p(y | X, \nu) \left( 1 + \log \frac{p(y | X, \theta)}{p(y | X, \nu)} \right) \\
&= p_\nu(y | X, \theta) \ .
\end{aligned}
$$

The second line stems from the inequality $t \geq 1 + \log t$.

$p_\nu(y|X, \theta) = p(y|X, \theta)$ is immediate when setting $\theta = \nu$ in Eq. 2.1. Deriving $p_\nu(y|X, \theta)$ with respect to $\theta$ yields

$$\frac{\partial p_\nu(y|X, \theta)}{\partial \theta} = p(y|X, \nu) \frac{\partial \log p(y|X, \theta)}{\partial \theta}$$
$$= \frac{p(y|X, \nu)}{p(y|X, \theta)} \frac{\partial p(y|X, \theta)}{\partial \theta} .$$

Taking $\theta = \nu$ on both sides yields $\left. \frac{\partial p_\nu(y|X,\theta)}{\partial \theta} \right|_{\nu=\theta} = \frac{\partial p(y|X,\theta)}{\partial \theta}$. □

Lemma 1 suggests that, if the current set of parameters is $\theta_t$, an appropriate upper bound on the probability that an example will be correctly classified is

$$L(\theta) = 1 - \frac{1}{N} \sum_i p(y_i|X_i, \theta)$$
$$\leq 1 - \frac{1}{N} \sum_i p(y_i|X_i, \theta_t) \left( 1 + \log \frac{p(y_i|X_i, \theta)}{p(y_i|X_i, \theta_t)} \right)$$
$$= C - \frac{1}{N} \sum_i p(y_i|X_i, \theta_t) \log p(y_i|X_i, \theta) ,$$

where $C$ is a constant independent of $\theta$. We shall denote

$$L_{\theta_t}(\theta) = -\frac{1}{N} \sum_i p(y_i|X_i, \theta_t) \log p(y_i|X_i, \theta) . \tag{2.5}$$

One possibility is to recompute the bound after every gradient step. This is exactly equivalent to directly minimizing $L$. Such a procedure is brittle. In particular, Eq. 2.5 indicates that, if an example is poorly classified early on, its gradient will be close to 0 and it will difficult to recover from this situation. Thus, we propose using Algorithm 1 for supervised learning: In regularly recomputing

**The data:** A dataset $\mathcal{D}$ comprising of $(X_i, y_i)$ pairs, initial parameters $\theta_0$
**The result:** Final parameters $\theta_T$
**for** *t = 0* **to** *T-1* **do**
 | $\theta_{t+1} = \arg\min_\theta L_{\theta_t} = -\sum_i p(y_i|X_i, \theta_t) \log p(y_i|X_i, \theta)$
**end**

**Algorithm 1:** Iterative supervised learning

the bound, we ensure that it remains close to the quantity we are interested in and that we do not waste time optimizing a loose bound.

The idea of computing tighter bounds during optimization is not new. In particular, several authors used a CCCP-based (Yuille & Rangarajan, 2003) procedure to achieve tighter bounds for SVMs (Xu et al., 2006; Collobert et al., 2006; Ertekin et al., 2011). Though Collobert et al. (2006) show a small improvement of the test error, the primary goal was to reduce the number of support vectors to keep the testing time manageable. Also, the algorithm proposed by Ertekin et al. (2011) required the setting of an hyperparameter, $s$, which has a strong influence on the final solution (see Fig. 5 in their paper). Finally, we are not aware of similar ideas in the context of the logistic loss.

Additionally, our idea extends naturally to the case where $p$ is a complicated function of $\theta$ and not easily written as a sum of a convex and a concave function. This might lead to nonconvex inner optimizations but we believe that this can still yield lower classification error. A longer study in the case of deep networks is planned.

REGULARIZATION

As this model further optimizes the training classification accuracy, regularization is often needed. The standard optimization procedure minimizes the following regularized objective:

$$\theta^* = \arg\min_\theta - \sum_i \log p(y_i|X_i, \theta) + \lambda\Omega(\theta)$$

$$= \arg\min_\theta - \sum_i \frac{1}{K} \log p(y_i|X_i, \theta) + \frac{\lambda}{K}\Omega(\theta) \ .$$

Thus, we can view this as an upper bound of the following "true" objective:

$$\theta^* = \arg\min_\theta - \sum_i p(y_i|X_i, \theta) + \frac{\lambda}{K}\Omega(\theta) \ ,$$

which can then be optimized using Algorithm 1.

ONLINE LEARNING

Because of its iterative nature, Algorithm 1 is adapted to a batch setting. However, in many cases, we have access to a stream of data and we cannot recompute the importance weights on all the points. A natural way around this problem is to select a parameter vector $\theta$ and to use $\nu = \theta$ for the subsequent examples. One can see this as "crystallizing" the current solution as the value of $\nu$ chosen will affect all subsequent gradients.

## 3 EXPERIMENTS

We experimented the impact of using tighter bounds to the expected misclassification rate on several datasets, which will each be described in their own section. The experimental setup for all datasets was as follows. We first set aside part of the dataset to compose the test set. We then performed k-fold cross-validation, using a generalized linear model, on the remaining datapoints for different values of $T$, the number of times the importance weights were recomputed, and the $\ell_2$-regularizer $\lambda$. For each value of $T$, we then selected the set of hyperparameters ($\lambda$ and the number of iterations) which achieved the lowest validation classification error. We computed the test error for each of the $k$ models (one per fold) with these hyperparameters. This allowed us to get a confidence intervals on the test error, where the random variable is the training set but not the test set.

For a fair comparison, each internal optimization was run for $Z$ updates so that $ZT$ was constant. Each update was computed on a randomly chosen minibatch of 50 datapoints using the SAG algorithm (Le Roux et al., 2012). Since we used a generalized linear model, each internal optimization was convex and thus had no optimization hyperparameter.

Fig. 1 presents the training classification errors on all the datasets.

### 3.1 COVERTYPE BINARY DATASET

The Covertype binary dataset (Collobert et al., 2002) has 581012 datapoints in dimension 54 and 2 classes. We used the first 90% for the cross-validation and the last 10% for testing. Due to the small dimension of the input, linear models strongly underfit, a regime in which tighter bounds are most beneficial. We see in Fig. 2 that using $T > 1$ leads to much lower training and validation classification errors. Training and validation curves are presented in Fig. 2 and the test classification error is listed in Table 1.

### 3.2 ALPHA DATASET

The Alpha dataset is a binary classification dataset used in the Pascal Large-Scale challenge and contains 500000 samples in dimension 500. We used the first 400000 examples for the cross-validation and the last 100000 for testing. A logistic regression trained on this dataset overfits quickly and, as a result, the results for all values of $T$ are equivalent. Training and validation curves are presented in Fig. 3 and the test classification error is listed in Table 2.

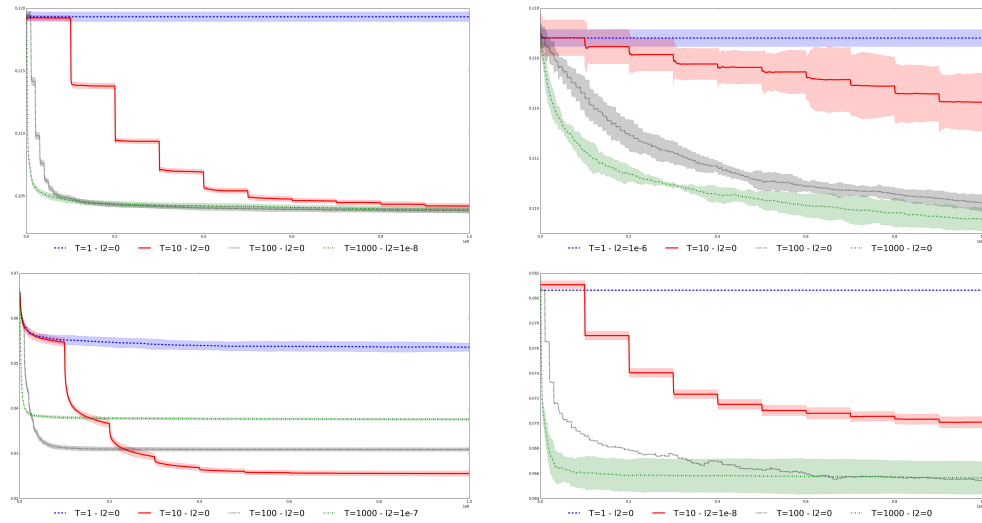

Figure 1: Training classification errors for *covertype* (**top left**), *alpha* (**top right**), *MNist* (**bottom left**) and *IJCNN* (**bottom right**). We can immediately see that all values of $T > 1$ yield significant lower errors than the standard log-loss (the confidence intervals represent $\pm$ 3 standard deviations).

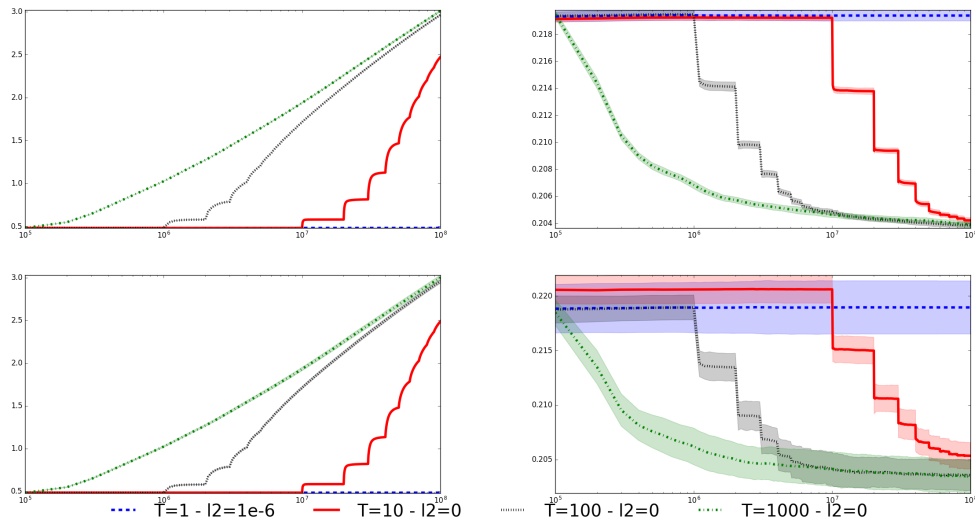

Figure 2: Training (**top**) and validation (**bottom**) negative log-likelihood (**left**) and classification error (**right**) for the *covertype* dataset. We only display the result for the value of $\lambda$ yielding the lowest validation error. As soon as the importance weights are recomputed, the NLL increases and the classification error decreases (the confidence intervals represent $\pm$ 3 standard deviations).

| T | $Z$ | Test error $\pm 3\sigma$ (%) |
|---|---|---|
| 1000 | 1e5 | $32.88 \pm 0.07$ |
| 100 | 1e6 | $32.96 \pm 0.06$ |
| 10 | 1e7 | $32.85 \pm 0.06$ |
| 1 | 1e8 | $36.32 \pm 0.06$ |

Table 1: Test error for the models reaching the best validation error for various values of $T$ on the *covertype* dataset. We can see that any value of $T$ greater than 1 leads to a significant improvement over the standard log-loss (the confidence intervals represent $\pm$ 3 standard deviations).

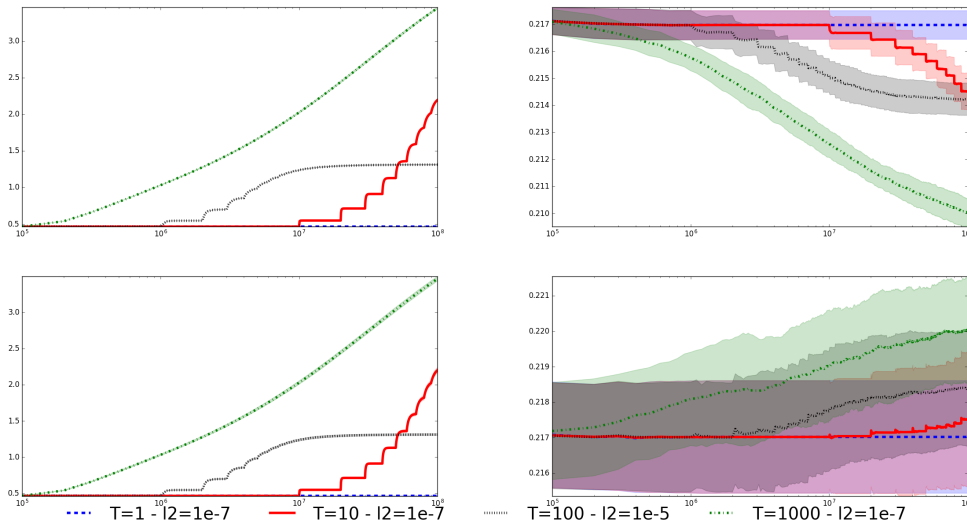

Figure 3: Training (**top**) and validation (**bottom**) negative log-likelihood (**left**) and classification error (**right**) for the *alpha* dataset. We only display the result for the value of $\lambda$ yielding the lowest validation error. As soon as the importance weights are recomputed, the NLL increases. Overfitting occurs very quickly and the best validation error is the same for all values of $T$ (the confidence intervals represent $\pm$ 3 standard deviations).

| T | Z | Test error $\pm3\sigma$ (%) |
|------|-----|-----------------------------|
| 1000 | 1e5 | $21.83 \pm 0.03$ |
| 100 | 1e6 | $21.83 \pm 0.03$ |
| 10 | 1e7 | $21.82 \pm 0.03$ |
| 1 | 1e8 | $21.82 \pm 0.03$ |

Table 2: Test error for the models reaching the best validation error for various values of $T$ on the *alpha* dataset. We can see that overfitting occurs very quickly and, as a result, all values of $T$ lead to the same result as the standard logloss.

## 3.3 MNIST DATASET

The MNist dataset is a digit recognition dataset with 70000 samples. The first 60000 were used for the cross-validation and the last 10000 for testing. Inputs have dimension 784 but 67 of them are always equal to 0. Despite overfitting occurring quickly, values of $T$ greater than 1 yield significant improvements over the log-loss. Training and validation curves are presented in Fig. 4 and the test classification error is listed in Table 3.

| T | Z | Test error $\pm3\sigma$ (%) |
|------|-----|-----------------------------|
| 1000 | 1e5 | $7.00 \pm 0.08$ |
| 100 | 1e6 | $7.01 \pm 0.05$ |
| 10 | 1e7 | $6.97 \pm 0.08$ |
| 1 | 1e8 | $7.46 \pm 0.11$ |

Table 3: Test error for the models reaching the best validation error for various values of $T$ on the *MNist* dataset. The results for all values of $T$ strictly greater than 1 are comparable and significantly better than for $T = 1$.

## 3.4 IJCNN DATASET

The IJCNN dataset is a dataset with 191681 samples. The first 80% of the dataset were used for training and validation (70% for training, 10% for validation, using random splits), and the last 20% were used for testing samples. Inputs have dimension 23, which means we are likely to be in the underfitting regime. Indeed, larger values of $T$ lead to significant improvements over the log-loss. Training and validation curves are presented in Fig. 5 and the test classification error is listed in Table 4.

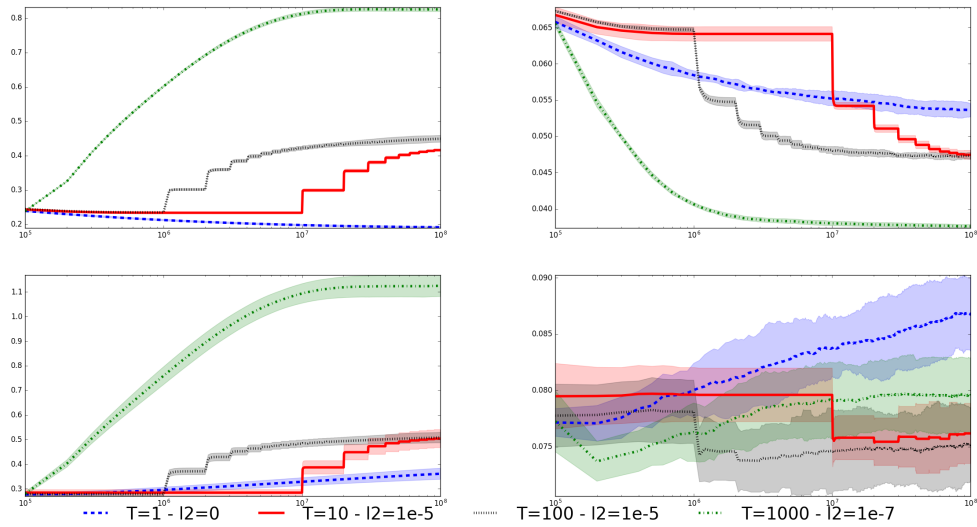

Figure 4: Training (**top**) and validation (**bottom**) negative log-likelihood (**left**) and classification error (**right**) for the *MNist* dataset. We only display the result for the value of $\lambda$ yielding the lowest validation error. As soon as the importance weights are recomputed, the NLL increases. Overfitting occurs quickly but higher values of $T$ still lead to lower validation error. The best training error was 2.52% with $T = 10$.

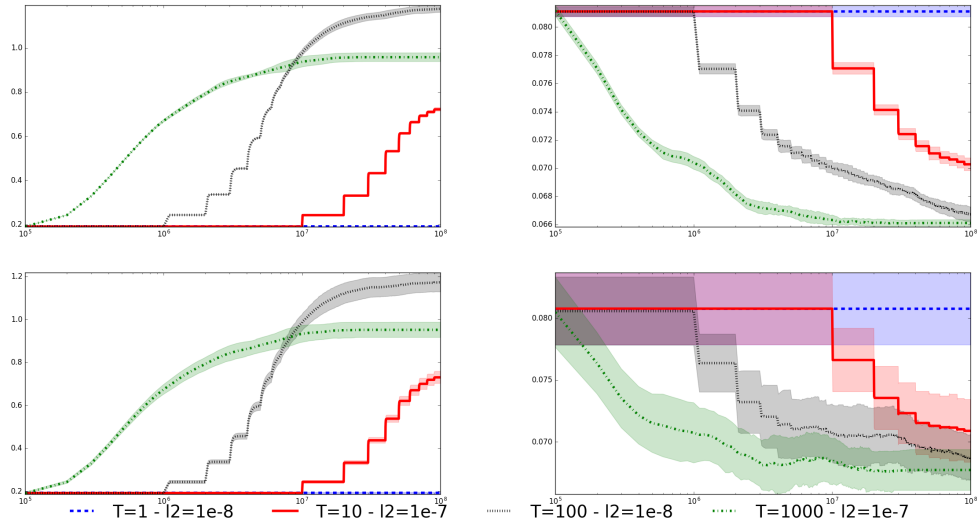

Figure 5: Training (**top**) and validation (**bottom**) negative log-likelihood (**left**) and classification error (**right**) for the *IJCNN* dataset. We only display the result for the value of $\lambda$ yielding the lowest validation error. As soon as the importance weights are recomputed, the NLL increases. Since the number of training samples is large compared to the dimension of the input, the standard logistic regression is underfitting and higher values of $T$ lead to better validation errors.

| T | Z | Test error $\pm 3\sigma$ (%) |
|------|-----|------------------|
| 1000 | 1e5 | $4.62 \pm 0.12$ |
| 100 | 1e6 | $5.26 \pm 0.33$ |
| 10 | 1e7 | $5.87 \pm 0.13$ |
| 1 | 1e8 | $6.19 \pm 0.12$ |

Table 4: Test error for the models reaching the best validation error for various values of $T$ on the *IJCNN* dataset. Larger values of $T$ lead significantly lower test errors.

# 4  SUPERVISED LEARNING AS POLICY OPTIMIZATION

We now propose an interpretation of supervised learning which closely matches that of direct policy optimization in reinforcement learning. This allows us to naturally address common issues in the literature, such as optimizing ROC curves or allowing a classifier to withhold taking a decision.

A machine learning algorithm is often only one component of a larger system whose role is to make decisions, whether it is choosing which ad to display or deciding if a patient needs a specific treatment. Some of these systems also involve humans. Such systems are complex to optimize and it is often appealing to split them into smaller components which are optimized independently. However, such splits might lead to poor decisions, even when each component is carefully optimized (Bottou). This issue can be alleviated by making each component optimize the full system with respect to its own parameters. Doing so requires taking into account the reaction of the other components in the system to the changes made, which cannot in general be modeled. However, one may cast it as a reinforcement learning problem where the environment is represented by everything outside of our component, including the other components of the system (Bottou et al., 2013).

Pushing the analogy further, we see that in one-step policy learning, we try to find a policy $p(y|X, \theta)$ over actions $y$ given the state $X$ [b] to minimize the expected loss defined as

$$\bar{L}(\theta) = -\sum_i \sum_y R(y, X_i) p(y|X_i, \theta) \ . \tag{4.1}$$

$\bar{L}(\theta)$ is equivalent to $L(\theta)$ from Eq. 1.3 where all actions have a reward of 0 except for the action choosing the correct class $y_i$ yielding $R(y_i, X_i) = 1$. One major difference between policy learning and supervised learning is that, in policy learning, we only observe the reward for the actions we have taken, while in supervised learning, the reward for all the actions is known.

Casting the classification problem as a specific policy learning problem yields a loss function commensurate with a reward. In particular, it allows us to explicit the rewards associated with each decision, which was difficult with Eq. 1.1. We will now review several possibilities opened by this formulation.

## OPTIMIZING THE ROC CURVE

In some scenarios, we might be interested in other performance metrics than the average classification error. In search advertising, for instance, we are often interested in maximizing the precision at a given recall. Mozer et al. (2001) address the problem by emphasizing the training points whose output is within a certain interval. Gasso et al. (2011); Parambath et al. (2014), on the other hand, assign a different cost to type I and type II errors, learning which values lead to the desired false positive rate. Finally, Bach et al. (2006) propose a procedure to find the optimal solution for all costs efficiently in the context of SVMs and showed that the resulting models are not the optimal models in the class.

To test the impact of optimizing the probabilities rather than a surrogate loss, we reproduced the binary problem of Bach et al. (2006). We computed the average training and testing performance over 10 splits. An example of the training set and the results are presented in Fig. 6.

Even though working directly with probabilities solved the non-concavity issue, we still had to explore all possible cost asymmetries to draw this curve. In particular, if we had been asked to maximize the true positive rate for a given false positive rate, we would have needed to draw the whole curve then find the appropriate point.

However, expressing the loss directly as a function of the probabilities of choosing each class allows us to cast this requirement as a constraint and solve the following constrained optimization problem:

$$\theta^* = \arg\min_\theta -\frac{1}{N_1} \sum_{i/y_i=1} p(1|x_i, \theta) \text{ such that } \frac{1}{N_0} \sum_{i/y_i=0} p(1|x_i, \theta) \leq c_{FP} \ ,$$

---

[b]In standard policy learning, we actually consider full rollouts which include not only actions but also state changes due to these actions.

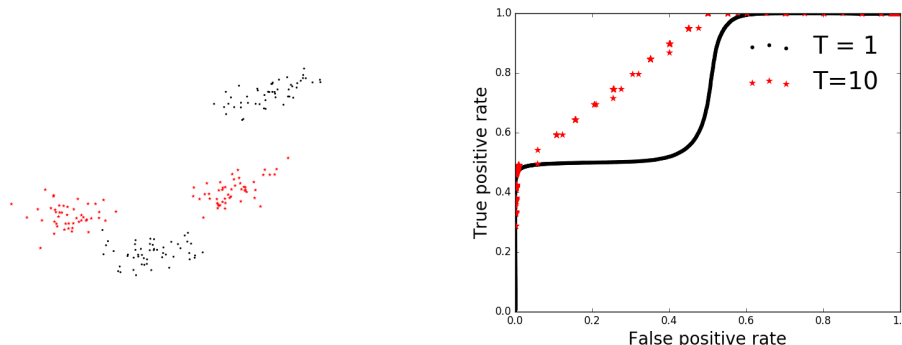

Figure 6: Training data (**left**) and test ROC curve (**right**) for the binary classification problem from Bach et al. (2006). The black dots are obtained when minimizing the log-loss for various values of the cost asymmetry. The red stars correspond to the ROC curve obtained when directly optimizing the probabilities. While the former is not concave, a problem already mentioned by Bach et al. (2006), the latter is.

with $N_0$ (resp. $N_1$) the number of examples belonging to class 0 (resp. class 1). Since $p(1|x_i, \theta) = 1 - p(0|x_i, \theta)$, we can solve the following Lagrangian problem

$$\min_{\theta} \max_{\lambda \geq 0} L(\theta, \lambda) = \min_{\theta} \max_{\lambda \geq 0} -\frac{1}{N_1} \sum_{i/y_i=1} p(1|x_i, \theta) + \lambda \left( 1 - \frac{1}{N_0} \sum_{i/y_i=0} p(0|x_i, \theta) - c_{FP} \right).$$

This is an approach proposed by Mozer et al. (2001) who then minimize this function directly. We can however replace $L(\theta, \lambda)$ with the following upper bound:

$$L(\theta, \lambda) \leq -\frac{1}{N_1} \sum_{i/y_i=1} p(1|x_i, \nu) \left( 1 + \log \frac{p(1|x_i, \theta)}{p(1|x_i, \nu)} \right)$$

$$+ \lambda \left( 1 - \frac{1}{N_0} \sum_{i/y_i=0} p(0|x_i, \nu) \left( 1 + \log \frac{p(0|x_i, \theta)}{p(0|x_i, \nu)} \right) - c_{FP} \right)$$

and jointly optimize over $\theta$ and $\lambda$. Even though the constraint is on the upper bound and thus will not be exactly satisfied during the optimization, the increasing tightness of the bound with the convergence will lead to a satisfied constraint at the end of the optimization. We show in Fig. 7 the obtained false positive rate as a function of the required false positive rate and see that the constraint is close to being perfectly satisfied. One must note, however, that the ROC curve obtained using the constrained optimization problems matches that of $T = 1$, i.e. is not concave. We do not have an explanation as to why the behaviour is not the same when solving the constrained optimization problem and when optimizing an asymmetric cost for all values of the asymmetry.

ALLOWING UNCERTAINTY IN THE DECISION

Let us consider a cancer detection algorithm which would automatically classify patients in two categories: healthy or ill. In practice, this algorithm will not be completely accurate and, given the high price of a misclassification, we would like to include the possibility for the algorithm to hand over the decision to the practitioner. In other words, it needs to include the possibility of being "Undecided".

The standard way of handling this situation is to manually set a threshold on the output of the classifier and, should the maximum score across all classes be below that threshold, deem the example too hard to classify. However, it is generally not obvious how to set the value of that threshold nor how it relates to the quantity we care about, even though some authors provided guidelines (**?**). The difficulty is heightened when the prior probabilities of each class are very different.

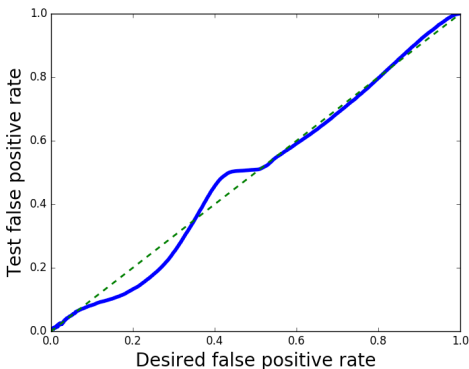

Figure 7: Test false positive rate as a function of the desired false positive rate $c_{FP}$. The dotted line representing the optimal behaviour, we can see that the constraint is close to being satisfied. $T = 10$ was used.

Eq. 4.1 allows us to naturally include an extra "action", the "Undecided" action, which has its own reward. This reward should be equal to the reward of choosing the correct class (i.e., 1) minus the cost $c_h$ of resorting to external intervention [c], which is less than 1 since we would otherwise rather have an error than be undecided. Let us denote by $r_h = 1 - c_h$ the reward obtained when the model chooses the "Undecided" class. Then, the reward obtained when the input is $X_i$ is:

$$R(y_i|X_i) = 1$$
$$R(``Undecided''|X_i) = r_h \ ,$$

and the average under the policy is $p(y_i|X_i, \theta) + r_h p(``Undecided''|X_i, \theta)$.

Learning this model on a training set is equivalent to minimizing the following quantity:

$$\theta^* = \arg\min_\theta -\frac{1}{N}\sum_i \left(p(y_i|X_i, \theta) + r_h p(``\text{Undecided}''|X_i, \theta)\right) \ . \tag{4.2}$$

For each training example, we have added another example with importance weight $r_h$ and class "Undecided". If we were to solve this problem through a minimization of the log-loss, it is well-known that the optimal solution would be, for each example $X_i$, to predict $y_i$ with probability $1/(1 + r_h)$ and "Undecided" with probability $r_h/(1 + r_h)$. However, when optimizing the weighted sum of probabilities, the optimal solution is still to predict $y_i$ with probability 1. In other words, adding the "Undecided" class does not change the model if it has enough capacity to learn the training set accurately.

## 5 Discussion and Conclusion

Using a general class of upper bounds of the expected classification error, we showed how a sequence of minimizations could lead to reduced classification error rates. However, there are still a lot of questions to be answered. As using $T > 1$ increases overfitting, one might wonder whether the standard regularizers are still adapted. Also, current state-of-the-art models, especially in image classification, already use strong regularizers such as dropout. The question remains whether using $T > 1$ with these models would lead to an improvement.

Additionally, it makes less and less sense to think of machine learning models in isolation. They are increasingly often part of large systems and one must think of the proper way of optimizing them in this setting. The modification proposed here led to an explicit formulation for the true impact of a classifier. This facilitates the optimization of such a classifier in the context of a larger production system where additional costs and constraints may be readily incorporated. We believe this is a critical venue of research to be explored further.

---

[c]This is assuming that the external intervention always leads to the correct decision. Any other setting can easily be used.

ACKNOWLEDGMENTS

We thank Francis Bach, Léon Bottou, Guillaume Obozinski, and Vianney Perchet for helpful discussions.

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
