# Peer review of "Tighter bounds lead to improved classifiers"

_ICLR 2017 — accepted_

[Official Review · AnonReviewer3 · rating 6 · confidence 4 · 19 Dec 2016 (modified: 29 Jan 2017)]
**Review & questions**

The paper proposes new bounds on the misclassification error. The bounds lead to training classifiers with an adaptive loss function, and the algorithm operates in successive steps: the parameters are trained by minimizing the log-loss weighted by the probability of the observed class as given by the parameters of the previous steps. The bound improves on standard log-likelihood when outliers/underfitting prevents the learning algorithm to properly optimize the true classification error. Experiments are performed to confirm the therotical intuition and motivation. They show different cases where the new algorithm leads to improved classification error because underfitting occurs when using standard log-loss, and other cases where the new bounds do not lead to any improvement because the log-loss is sufficient to fit the dataset.

The paper also discusses the relationship between the proposed idea and reinforcement learning, as well as with classifiers that have an "uncertain" label. 

While the paper is easy to read and well-written overall, in a second read I found it difficult to fully understand because two problems are somewhat mixed together (here considering only binary classification for simplicity): 
(a) the optimization of the classification error of a *randomized* classifier, which predicts 1 with probability P(1|x, theta), and 
(b) the optimization of the deterministic classifier, which predicts sign(P(1|x, theta) - 0.5), in a way that is robust to outliers/underfitting. 

The reason why I am confused is that "The standard approach to supervised classification", as is mentioned in the abstract, is to use deterministic classifiers at test time, and the log-loss (up to constants) is an upper bound on the classification error of the deterministic classifier. However, the bounds discussed in the paper only concern the randomized classifier.

=== question:
In the experiments, what kind of classifier is used? The randomized one (as would the sentence in the first page suggest "Assuming the class is chosen according to p(y|X, θ)"), or the more standard deterministic classifier argmax_y P(y|x, theta) ?

As far as I can see, there are two cases: either (i) the paper deals with learning randomized classifiers, in which case it should compare the performances with the deterministic counterparts that people use in practice, or (ii) the paper makes sense as soon as we accept that the optimization of criterion (a) is a good surrogate for (b). In both cases,  I think the write-up should be made clearer (because in case (ii) the algorithm does not minimize an upper bound on the classification error, and in case (i) what is done does not correspond to what is usually done in binary classification). 

=== comments:
- The section "allowing uncertainty in the decision" may be improved by adding some references, e.g. Bartlett & Wegkamp (2008) "Classification with a Reject Option using a Hinge Loss" or Sayedi et al. (2010) "Trading off Mistakes and Don’t Know Predictions".

- there seems to be a "-" sign missing in the P(1|x, theta) in L(theta, lambda) in Section 3.

- The idea presented in the paper is interesting and original. While I give a relatively low score for now, I am willing to increase this score if the clarifications are made.

Final comments:
I think the paper is clear enough in its current form, even though there should still be improvement in the justification of why and to what extent the error of the randomized classifier is a good surrogate for the error of the true classifier. While the "smoothed" version of the 0/1 loss is an acceptable explanation in the standard classification setup, it is less clear in the section dealing with an additional "uncertain" label. I increase my score from 5 to 6.

[Official Review · AnonReviewer2 · rating 4 · confidence 4 · 19 Dec 2016]
**No Title**

The paper proposes an alternative to conditional max. log likelihood for training discriminative classifiers. The argument is that the conditional log. likelihood is an upper bound of the Bayes error which becomes lousy during training. The paper then proposes better bounds computed and optimized in an iterative algorithm. Extensions of this idea are developed for regularized losses and a weak form of policy learning. Tests are performed on different datasets.

An interesting aspect of the contribution is to revisit a well-accepted methodology for training classifiers. The idea looks fine and some of the results seem to validate it. This is however still a preliminary work and one would like to see the ideas pushed further. Globally, the paper lacks coherence and depth: the part on policy learning is not well connected to the rest of the paper and the link with RL is not motivated in the two examples (ROC optimization and uncertainties). The experimental part needs a rewriting, e.g. I did not find a legend for identifying the different curves in the figures, which makes difficult to appreciate the results.

[Official Review · AnonReviewer1 · rating 8 · confidence 5 · 20 Dec 2016 (modified: 24 Jan 2017)]

The paper analyses the misclassification error of discriminators and highlights the fact that while uniform probability prior of the classes makes sense early in the optimization, the distribution deviates from this prior significantly as the parameters move away from the initial values. 
Consequently, the optimized upper bound (log-loss) gets looser. 

As a fix, an optimization procedure based on recomputing the bound is proposed. The paper is well written. While the main observation made in this paper is a well-known fact, it is presented in a clear and refreshing way that may make it useful to a wide audience at this venue. 

I would like to draw the author's attention to the close connections of this framework with curriculum learning. More on this can be found in [1] (which is a relevant reference that should be cited). A discussion on this could enrich the quality of the paper. 

There is a large body of work on directly optimizing task losses[2][3] and the references therein. These should also be discussed and related particularly to section 3 (optimizing the ROC curve).

[1] Training Highly Multiclass Classifiers, Gupta et al. 2014.
[2] Direct Loss Minimization for Structured Prediction, McAllester et al. 
[3] Generalization Bounds and Consistency for Latent Structural Probit and Ramp Loss, McAllester and Keshet.

Final comment:
I believe the material presented in this paper is of interest to a wide audience at ICLR.
The problem studied is interesting and the proposed approach is sound. 
I recommend to accept the paper and increase my score (from 7 to 8).

[Final Decision · Program Chairs · 06 Feb 2017]
**ICLR committee final decision**

This is a well written paper that proposes the adaptation of the loss function for training during optimization, based on a simple and effective tighter bound on classification error. The paper could be improved in terms of a) review of related works; b) more convincing experiments.